# Certified Robustness of Graph Convolution Networks for Graph Classification under Topological Attacks

**Hongwei Jin**∗, **Zhan Shi**∗, **Ashish Peruri**, **Xinhua Zhang**
Department of Computer Science
University of Illinois at Chicago
Chicago, IL 60607
{hjin25,zshi22,vperur2,zhangx}@uic.edu

## Abstract

Graph convolution networks (GCNs) have become effective models for graph classification. Similar to many deep networks, GCNs are vulnerable to adversarial attacks on graph topology and node attributes. Recently, a number of effective attack and defense algorithms have been designed, but no certificate of robustness has been developed for GCN-based graph classification under *topological perturbations* with both local and global budgets. In this paper, we propose the first certificate for this problem. Our method is based on Lagrange dualization and convex envelope, which result in tight approximation bounds that are efficiently computable by dynamic programming. When used in conjunction with robust training, it allows an increased number of graphs to be certified as robust.

## 1 Introduction

Graph convolution networks [GCNs, 1] have been shown very effective for modeling graph structured data such as social networks [2] and protein interactive networks [3]. In this paper, we focus on the application of GCN to graph classification, where given an attributed graph, the task is to predict its class label. This is different from node classification where each node is assigned a (possibly different) label. Along with other models such as graph kernels [4, 5], GCNs have achieved strong performance in this task [6–9].

However, similar to most deep learning models, GCNs are also vulnerable to adversarial attacks that maliciously perturb the data to induce specific errors [10, 11]. These include node attacks (*i.e.*, perturbing node features) [12–14] and topological attacks (*i.e.*, adding or removing edges) [11, 15]. They pose serious challenges because adversaries are often omnipresent in their typical application scenarios [16–18], and even when no manual or computerized adversary is present, it is still important to analyze the worst-case robustness of the model by treating nature as an adversary.

Since the strongest attack is often intractable to compute, an alternative approach that has recently attracted considerable interest is to construct certificates of robustness, *i.e.*, certain sufficient conditions which, once satisfied, guarantee the immunity to *any* admissible attack (see Related Work). Unfortunately such results have been scarce on graph data. [12] certified GCNs against node feature perturbations for node classification. Under topological perturbation, [19] developed certificates for PageRank and label/feature propagation, while [20] developed them for community detection.

Unfortunately, all of these techniques meet with substantial obstacles or slackness when extended to GCNs for graph classification under topological attacks, an important application setting where vulnerability is a real issue (Section 5). First of all, the technique of restricting to the $k$-hop subgraph in node classification such as [12, 21] can no longer be utilized because graph classification requires

---

∗Equal contribution

pooling together the hidden representation of *all* nodes. Secondly, GCNs have a specific normalization structure which introduces a new type of nonlinearity; treating it as a generic function leads to loose certificates. Thirdly, the set of admissible topological perturbations (*a.k.a.* threat model) may have a refined structure. For example, besides the global budgets, the number of perturbations *per node* is often subjected to a local budget. This is not considered by [20] or [22], and extending them to such a setting is nontrivial. [19] does not enforce symmetry on the attacked affinity matrix.

Our certificate is the first to address all these challenges. Given a trained GCN and a threat model with both local and global budgets, it can efficiently verify that no topological perturbation can change the graph prediction. Our main tool is dualization (Section 3) and convex envelope (Section 4), with the latter providing the *tightest* convex lower bound on a refined threat domain [23], and it can be computed efficiently through dynamic programming. As a byproduct, we also developed a new attack algorithm (Section 3.2), which, when used in conjunction with the certificate, confirms empirically that both the attack and the certificate are often tight (Section 5).

**Related Work**    We first overview the existing techniques of robust certificate, shedding light on the obstacles in directly extending them to GCNs for graph classification under topological attacks. A survey of adversarial attack and defence for graph structured data is available at [24]. Most related to ours is the convex envelope relaxation of the ReLU activation [25], which has been generalized by [26–29]. However, GCNs present a new source of nonlinearity left to be addressed: normalization in the graph Laplacian. Methods based on curvature or Lipschitz continuous constant enjoy more generality [30–32], but they depend on estimating the local or global curvature or engineering Lipschitz layers, both of which are complicated by the discrete domain. It is also hard to quantify the slackness in these relaxations, a problem that has been mitigated by our approach because the convex envelope of the margin provides the tightest convex lower approximation.

Randomized smoothing adds noise to the input [33–35], and it has been extended to discrete noises [20, 22]. However, the presence of both local *and* global budgets makes it hard to design the noise distribution and to analyze the certificate. Semi-definite relaxation often leads to loose bounds [36, 37].

## 2   Preliminaries

We consider the task of graph classification. The training set consists of pairs of $(G_k, y_k)$, where $G_k$ is a directed or undirected graph, and $y_k \in [C] := \{1, \dots, C\}$ is its label for multiclass classification. Given a graph $G$, its vertices are denoted as $1, 2, \dots, n$, where $n$ is the number of nodes in $G$. We represent the topology of $G$ by its adjacency matrix $A$, where $A_{ij} = 1$ if $i \to j$ is an edge, and is 0 otherwise. $A$ is symmetric if the graph is undirected. For simplicity, we set $A_{ii} = 0$, i.e., no self-loop, although our method can be easily extended to accommodate self-loops.

Each node also has its own features $x_i \in \mathbb{R}^d$, and we stack them into a matrix $X := (x_1, \dots, x_n)^\top \in \mathbb{R}^{n \times d}$. So a graph $G$ is uniquely characterized by the tuple $(A, X)$. The graph convolution network (GCN) mixes the hidden representation of nodes through a weight matrix $\hat{D}^{-\frac{1}{2}} \hat{A} \hat{D}^{-\frac{1}{2}}$, where $\hat{A} = A + I$ with $I$ being the identity matrix, and $\hat{D}$ is a diagonal matrix whose diagonal is $\hat{A}\mathbf{1}$ ($\mathbf{1}$ is a vector of all ones). For simplicity, it is also common to use $\hat{D}^{-1}\hat{A}$ instead of $\hat{D}^{-\frac{1}{2}} \hat{A} \hat{D}^{-\frac{1}{2}}$ [*e.g.*, 19], and we will adopt $\hat{D}^{-1}\hat{A}$ in our development as it does not lead to significant difference in performance. We denote the $i$-th row and $i$-th column of $A$ as $A_{i:}$ and $A_{:i}$, respectively.

The one-layer GCN for graph classification learns a hidden-layer weight matrix $W \in \mathbb{R}^{d \times d'}$ and an output-layer weight matrix $U \in \mathbb{R}^{d' \times c}$, so that the following empirical risk is minimized:

$$\min_{W,U} \mathbb{E}_{(G,y) \sim \tilde{p}} \, \ell(\text{pooling}(\sigma(\hat{D}^{-1}\hat{A}XW))U, y). \tag{1}$$

Here $\tilde{p}$ is the empirical distribution of graph-label pairs, $\ell$ is a loss function, and $\sigma$ is the activation function, which is applied element-wise on a matrix. The pooling function aggregates the hidden representation of all nodes in a graph, and examples include i) averaging: $H \mapsto \frac{1}{n}\mathbf{1}^\top H$, where $H = \sigma(\hat{D}^{-1}\hat{A}XW)$; ii) maximum: $H \mapsto (\max(H_{:1}), \dots, \max(H_{:d'}))$; iii) attention: $H \mapsto \alpha^\top H$, where $\alpha_i \geq 0$ and $\mathbf{1}^\top \alpha = 1$. In this paper, we consider attention pooling where $\alpha$ is allowed to depend on $X$ but *not* on $A$. Obviously it subsumes average pooling with $\alpha = n^{-1}\mathbf{1}$.

Let $f$ be a (possibly nonconvex) function on a domain $\mathcal{M} \subseteq \mathbb{R}^m$. Then its Fenchel dual $f^*$ is defined as $f^*(z) = \sup_{x \in M}\{x^\top z - f(x)\}$, and the convex envelope $f^{**}$ is the Fenchel dual of $f^*$ [23]. Note that changing the domain $\mathcal{M}$ may change $f^*$. It is known that any closed convex function $g$ that minorizes $f$ (*i.e.*, $g(x) \leq f(x)$ for all $x \in \mathcal{M}$) must satisfy that $f^{**}(x) \geq g(x)$ for all $x \in \mathcal{M}$.

## 2.1 Threat model, margin, and robustness certificate

Our goal is to attack the learned model by perturbing the graph topology $A$ under the trained values of $W$ and $U$. It is also of interest to perturb $X$, but this is a continuous problem or has been well studied [12]. Attacking $A$, in contrast, is much more challenging because it leads to a discrete optimization problem under involved constraints: $\max_{A \in \mathcal{A}} \ell(\text{pooling}(\sigma(\hat{D}^{-1}\hat{A}XW))U, y)$, where $\mathcal{A}$ is the admissible set of perturbed graphs in $\{0, 1\}^{n \times n}$ with $A_{ii} = 0$. Examples of $\mathcal{A}$ include the following subsets of $\{0, 1\}^{n \times n}$ with $A_{ii} = 0$:

$\mathcal{A}^1$: $A$ can be asymmetric, but for each node, at most $\delta_l$ incident edges can be added or removed. Here $\delta_l$ is a small integer (*e.g.*, 3), which can also vary for different nodes $i$. Formally, we can write it as $\left\|A_{i:} - A_{i:}^{ori}\right\|_1 \leq \delta_l$, where $A^{ori}$ is the original graph from the dataset.

$\mathcal{A}^2$: $A$ can be asymmetric, but at most $2\delta_g$ directed edges (or $\delta_g$ undirected edges) can be added or removed across all nodes. We write it as $\left\|A - A^{ori}\right\|_1 := \sum_i \left\|A_{i:} - A_{i:}^{ori}\right\|_1 \leq 2\delta_g$.

More refined threat models that distinguish dropping edges from adding them can be easily accommodated in our framework too. Furthermore, we consider the symmetry constraint using $\mathcal{A}^3 := \{A \in \mathbb{R}^{n \times n} : A^\top = A\}$, which is useful for modeling undirected graphs. This is a convex set. It is conceivable that $\mathcal{A}^1$ is the simplest because the constraints are local, *i.e.*, decomposed over nodes. These three constraints can be combined/intersected, and we denote the result via the superscript, *e.g.*, $\mathcal{A}^{1+3} := \mathcal{A}^1 \cap \mathcal{A}^3$.

**Certificate.** Given $W$ and $U$, let $z_c = \text{pooling}(\sigma(\hat{D}^{-1}\hat{A}XW))U_{:c}$ be the discriminant value for class $c$. The worst-case margin with attention pooling is defined as

$$\min_{A \in \mathcal{A}} \min_c z_y - z_c \quad = \quad \min_c \min_{A \in \mathcal{A}} F_c(A), \tag{2}$$

$$\text{where} \quad F_c(A) := \sum_{i=1}^n \underbrace{\alpha_i \sigma\left((\hat{A}_{i:}\mathbf{1})^{-1}\hat{A}_{i:}XW\right)(U_{:y} - U_{:c})}_{=:f_{c,i}(A_{i:})}. \tag{3}$$

If the minimal value is positive, then the trained model is certifiably robust. It suffices to minimize $F_c(A)$ for each fixed value of $c \in [C]\backslash\{y\}$. In practice, any *lower bound* on the minimum value of $F_c(A)$ can serve as a **certificate of robustness**: if it is positive, then the model must be robust. On the other hand, any $A$ and $c$ that make $F_c(A)$ negative discloses its vulnerability as an attack. To lighten notation, we will henceforth suppress the subscript $c$, and just write $F(A)$ and $f_i(A_{i:})$.

Our certificate will be presented under one hidden layer. Different from other neural networks, GCNs usually do not benefit from more than two hidden layers (*i.e.*, applying the graph Laplacian twice) [1, 3, 11, 12]. Different from node classification which directly uses the output of the final convolution layer as node-wise classification logits, graph classification has additional weights $U$ to optimize, along with optimizable pooling weights. So in practice, two hidden layers are not essential. For completeness, we will detail the extension to multiple hidden layers in Appendix D.

## 3 Certifying Robustness by Lagrange Duality

Our first certificate is based on the weak duality in Lagrange dualization. It provides exact certificates (*i.e.*, exact attacks) for $\mathcal{A}^1$, $\mathcal{A}^2$, and $\mathcal{A}^{1+2}$ at the cost of $O(n^2)$ for linear activations and $O(n^{\delta_l})$ for any nonlinear activation. We emphasize that linear activation is *not* an over-simplification for GCNs, because it very often performs as well or even better than a nonlinear activation [38].

To start with, it is straightforward to minimize $F(A)$ over $\mathcal{A}^1$ and $\mathcal{A}^2$, leading to an exact certificate. Under $\mathcal{A}^1$, $F(A)$ can be decomposed over $A_{i:}$ across nodes $i$, allowing each $A_{i:}$ to be solved separately. When $\sigma$ is the linear activation (*i.e.*, identity function), each $A_{i:}$ in $f_i(A_{i:})$ can be solved

---
**Algorithm 1:** Dynamic programming for minimizing $F(A)$ under $\mathcal{A}^{1+2}$
---
**1** Initialize by $s_0(0) = 0$ and $c_0 = 0$. Precompute $a_i(j)$ for all $i \in [n]$ and $j \in [0, \delta_l]$.
**2 for** $i = 1, 2, \ldots, n$ **do**
**3** $\quad c_i \leftarrow \min\{c_{i-1} + \delta_l, 2\delta_g\}$.
**4** $\quad s_i(j) \leftarrow \min_k\{s_{i-1}(j-k) + a_i(k) : k \in [0, \delta_l], j - k \in [0, c_{i-1}]\}$ for all $j \in [0, c_i]$.
**5** Pick $j_n \leftarrow \arg\min_j s_n(j)$.
**6 for** $i = n, n-1, \ldots, 1$ **do**
**7** $\quad k_i \leftarrow \arg\min_k\{s_{i-1}(j_t - k) + a_i(k) : k \in [0, \delta_l], j_i - k \in [0, c_{i-1}]\}$ and $j_{i-1} \leftarrow j_i - k_i$.
**8** $\quad$ Recover the optimal $A_{i:}$ based on the argmin in the definition of $a_i(k_i)$ in (4).
**9 Return** $A$
---

by sorting, which costs $O(\delta_l + n \log n)$ computation. The details are in Appendix C.1, and the same strategy applies to $\mathcal{A}^2$. When $\sigma$ is not linear, we can enumerate all the possible selections of $\delta_l$ neighbors among the $n - 1$ ones, and that costs $O(\binom{n}{\delta_l})$ complexity, which is still polynomial in $n$ for a fixed (small) value of $\delta_l$. We will propose better solutions in Section 4.3 via convex envelope.

Using this technique, we can compute the exact attack for $\mathcal{A}^{1+2}$ at a cost of $O(n^2)$ for linear activation and $O(n^{\delta_l})$ for nonlinear activation. First apply the above technique for $\mathcal{A}^1$ to precompute

$$a_i(j) := \min\{f_i(A_{i:}) : A_{i:} \text{ satisfies } \left\|A_{i:} - A_{i:}^{ori}\right\|_1 = j\}, \quad \forall\, i \in [n],\, j \in [0, \delta_l]. \tag{4}$$

Then minimizing $F(A)$ over $\mathcal{A}^{1+2}$ is equivalent to minimizing $\sum_i a_i(k_i)$ subject to $k_i \in [0, \delta_l]$ and $\sum_{i=1}^n k_i \le 2\delta_g$. This can be solved by dynamic programming as shown in Algorithm 1, where the central quantity for update is $s_i(j)$ — the lowest possible value of $F(A)$ under $\sum_{i'=1}^i k_{i'} = j$ and $k_{i'} = 0$ for all $i' > i$. Clearly the computational cost for the dynamic programming is $O(n\delta_g\delta_l)$, and the storage cost is $O(n\delta_g)$. We must also factor in the cost of computing $\{a_i(j) : i \in [n],\, j \in [0, \delta_l]\}$. When $\sigma$ is identity, it costs $O(\delta_l^2 n + n^2 \log n)$. Otherwise, the above recipe costs $O(\binom{n}{\delta_l}n)$.

Unfortunately, polytime exact solutions are not available to $\mathcal{A}^{1+3}$, $\mathcal{A}^{2+3}$, and $\mathcal{A}^{1+2+3}$ even for linear activation. See some hardness results in [39]. We will next show how to solve them approximately.

## 3.1 Approximate certificates of robustness for $\mathcal{A}^{1+3}$, $\mathcal{A}^{2+3}$, and $\mathcal{A}^{1+2+3}$ via dualization

Since exact solutions are available only in the aforementioned asymmetric constraints, we will next develop a lower bound (*i.e.*, certificate of robustness) for $F(A)$ under $\mathcal{A}^{1+3}$, $\mathcal{A}^{2+3}$, and $\mathcal{A}^{1+2+3}$. Since exact solution is already available under $\mathcal{A}^{1+2}$, our focus will be on addressing the additional constraint $\mathcal{A}^3$. To this end, we will decompose $\mathcal{A}^{1+2+3}$ into $\mathcal{A}^{1+2} \cap \mathcal{A}^3$, and resort to weak duality:

$$\min_{A \in \mathcal{A}^{1+2+3}} F(A) = \min_{A \in \mathcal{A}^{1+2}} \max_\Lambda F(A) + \operatorname{tr}(\Lambda^\top (A - A^\top)) \tag{5}$$

$$\ge \max_\Lambda \min_{A \in \mathcal{A}^{1+2}} \left\{ F(A) + \operatorname{tr}((\Lambda^\top - \Lambda)A) \right\}. \tag{6}$$

Fixing $\Lambda$, the inner optimal solution for $A \in \mathcal{A}^{1+2}$ can be found in its *exact* value by Algorithm 1 for both linear and nonlinear activations, because $\operatorname{tr}((\Lambda^\top - \Lambda)A)$ is decomposed over the rows of $A$. With the optimal $A^*$, the supergradient in $\Lambda$ can be evaluated via Danskin's theorem: $A^* - (A^*)^\top$. This approach generalizes directly to $\mathcal{A}^{1+3}$ and $\mathcal{A}^{2+3}$: just use $\mathcal{A}^1$ and $\mathcal{A}^2$ in (6) respectively.

Due to lack of strong duality, the argmin of $A$ in (6) extracted under the optimal $\Lambda$ is not guaranteed to be optimal or even symmetric. That said, this approach is meant to provide a lower bound that certifies the robustness of the model; extracting a feasible $A$ itself is only a secondary pursuit.

## 3.2 Emprically characterizing the certificate's tightness: approximate attack by ADMM

Although a lower bound of $F(A)$ certifies robustness, it is still unclear how tight it is. This can be resolved by designing an *upper bound*. To this end, we propose the alternating direction method of multipliers (ADMM), which has been used extensively in convex and nonconvex optimization to address complicated constraints or objectives, where each component is simple enough to admit efficient or closed-form proximal mapping. We leverage this idea and extend it to a discrete setting by noting $\mathcal{A}^{1+2+3} = \mathcal{A}^{1+2} \cap \mathcal{A}^3$, so that Algorithm 1 for $\mathcal{A}^{1+2}$ can be utilized:

$$\min_{A,B} F(A) + \delta(A \in \mathcal{A}^{1+2}) + \delta(B \in \mathcal{A}^3), \quad s.t. \ A = B. \tag{7}$$

Here $\delta(\cdot) = 0$ if $\cdot$ is true, and $\infty$ otherwise. The ADMM algorithm, which is designed for continuous optimization, can be easily extended to our discrete setting with almost no change. First introduce the augmented Lagrangian with a small positive constant $\mu$ and Frobenious norm $\|\cdot\|_F$:

$$\mathcal{L}(A, B, \Lambda) := F(A) + \delta(A \in \mathcal{A}^{1+2}) + \delta(B \in \mathcal{A}^3) - \text{tr}(\Lambda^\top (A - B)) + \tfrac{1}{2\mu} \|A - B\|_F^2, \quad (8)$$

Then ADMM loops as follows with $B_0$ initialized to, *e.g.*, the optimal $A$ from the dual objective (6):

$$A_{t+1} \leftarrow \arg\min_A \mathcal{L}(A, B_t, \Lambda_t) = \arg\min_{A \in \mathcal{A}^{1+2}} F(A) - \text{tr}(\Lambda_t^\top A) + \tfrac{1}{2\mu} \|A - B_t\|_F^2 \quad (9)$$

$$B_{t+1} \leftarrow \arg\min_B \mathcal{L}(A_{t+1}, B, \Lambda_t) = \arg\min_{B \in \mathcal{A}^3} \left\{ \text{tr}(\Lambda_t^\top B) + \tfrac{1}{2\mu} \|A_{t+1} - B\|_F^2 \right\} \quad (10)$$

$$\Lambda_{t+1} \leftarrow \Lambda_t + \tfrac{1}{\mu}(B_{t+1} - A_{t+1}) \quad (11)$$

Step (9) is solvable by Algorithm 1 as $x^2 = x$ for $x \in \{0, 1\}$. (10) has a closed-form solution: $\frac{1}{2}(A_t + A_t^\top - \mu\Lambda_t - \mu\Lambda_t^\top)$. More discussions are deferred to Appendix B.

## 4 Certifying Robustness by Convex Envelope

The above dualization based method relies on the efficient computation of $a_i(j)$ in (4), which can be expensive for ReLU. In addition, although dualization provides a lower bound, it is not the tightest. In this section, we show that both of the issues can be resolved by approximating the feasible domain $\mathcal{A}$ with the smallest enclosing convex set (convex hull), and substituting the objective function with the tightest convex minorant, *i.e.*, the convex envelope [23]. Although convex optimization appears a "solved" problem, there can still be "hidden" NP-hardness which hinders their application to neural networks. Firstly, it is nontrivial to construct a convex function that is both efficiently computable and matches the value of $F(A)$ on $\mathcal{A}$. Furthermore, the convex hull of the domain may not be characterized by simple (generalized) inequalities, and even projecting to it can also be NP-hard.

As one of our major contributions, we will demonstrate that both of the obstacles can be overcome in a nontrivial fashion thanks to the amenable structures in $F(A)$ and $\mathcal{A}$. In particular, $F(A)$ can be extended from $\mathcal{A}$ to its convex hull $\text{co}\,\mathcal{A}$ as a convex function $F^\circ(Z)$ that matches $F$ on $\mathcal{A}$ (*i.e.*, $F^\circ(A) = F(A)$ for $A \in \mathcal{A}$), while retaining computational efficiency. As a result, the optimization

$$\min F^\circ(Z), \quad s.t. \quad Z \in \text{co}\,\mathcal{A}, \quad (12)$$

can be solved i) exactly over the convex hull of $\mathcal{A}^{1+2}$, $\mathcal{A}^{1+3}$, or $\mathcal{A}^{2+3}$; and ii) with $\frac{1}{2}$-approximation guarantee over the convex hull of $\mathcal{A}^{1+2+3}$. Although it is still challenging to quantify the gap to the original $\min_{A \in \mathcal{A}} F(A)$, the feasibility of using convex envelope is still intuitively pleasing because it provides the tightest convex minorant, and (12) does serve as a bona fide certificate of robustness.

**Overview of the optimization algorithm.** As will be seen later, although the explicit form of $\text{co}\,\mathcal{A}$ can be easily derived for $\mathcal{A}^1$ and $\mathcal{A}^2$, difficulties arise once $\mathcal{A}^1$ and $\mathcal{A}^2$ are intersected with $\mathcal{A}^3$. In these cases, projection to $\text{co}(\mathcal{A})$ turns out hard, and even explicitly expressing $\text{co}(\mathcal{A})$ in terms of linear constraints is nontrivial, precluding the application of projected gradient descent algorithm. To bypass this difficulty, we adopted the conditional gradient algorithm [CG, 40], which instead of projecting to $\text{co}(\mathcal{A})$, resorts to maximizing a *linear function* over $\text{co}(\mathcal{A})$ (polar operator). As a key benefit, it is equivalent to maximizing a linear function over $\mathcal{A}$, which turns out efficiently solvable as shown in Section 4.1.

In a nutshell, CG builds a first-order Tayler approximation of the objective function $F^\circ$ at the current solution $Z_t$, and then invokes the polar operator (PO) to find a corner of $\text{co}(\mathcal{A})$ (denoted as $B_t$) that minimizes this approximation over $\text{co}(\mathcal{A})$. Then $Z_t$ is updated by moving towards $B_t$ with a sensible step size. Whenever the PO is exactly computable, the final optimal solution to (12) can be found in $O(1/\epsilon)$ iterations. The CG algorithm is detailed as follows:

---
1. Initialize $Z_0$ to any arbitrary element in $\text{co}\,\mathcal{A}$ (or just in $\mathcal{A}$).
2. For $t = 1, 2, \ldots$
3. **PO**: find $B_t \in \arg\max_{Z \in \text{co}\,\mathcal{A}} \text{tr}(R^\top Z) = \arg\max_{A \in \mathcal{A}} \text{tr}(R^\top A)$, where $R = -\nabla F^\circ(Z_{t-1})$.
4. Update $Z_t \leftarrow (t+2)^{-1}(2B_t + tZ_{t-1})$.
---

Note that CG queries the domain only through the PO in Step 3. Sections 4.1 and 4.2 will respectively address the efficient computation of the PO and the gradient of $F^\circ(A)$.

**Algorithm 2:** Greedy algorithm for solving the polar operator under $\mathcal{A}_{\circ}^{1+2+3}$

---

**1** Initialize $V = \mathbf{0}$ and set $\hat{J} = (J + J^\top)/2$.
**2** Sort the indices $\mathcal{I} := \{(i,j) : j > i, \hat{J}_{ij} > 0\}$ in a descending order of $\hat{J}_{ij}$.
**3 for** $(i,j) \in \mathcal{I}$ *(in the sorted order)* **do**
**4** $\quad$ Set $\hat{V} = V$, followed by $\hat{V}_{ij} \leftarrow 1$ and $\hat{V}_{ji} \leftarrow 1$. If $\hat{V} \in \mathcal{A}^{1+2+3}$, then $V \leftarrow \hat{V}$.
**5 Return** $V$

---

## 4.1 Convexification of $\mathcal{A}$ and its polar operators

We first demonstrate how the PO in CG can be efficiently computed when $\mathcal{A}$ consists of the intersection of $\mathcal{A}^1$, $\mathcal{A}^2$, and/or $\mathcal{A}^3$. Let $V_{ij}$ encode whether $A_{ij}$ changes upon $A^{ori}$ (1 for yes and 0 for no). Then $A = \frac{1}{2}[(2A^{ori} - E) \circ (-2V + E) + E]$, where $\circ$ represents the elementwise product, and $E$ is a matrix of all ones. Now the PO problem can be converted to $\arg \max_{V \in \mathcal{A}_\circ} \text{tr}(J^\top V) + \text{tr}(R^\top A^{ori})$, where $\mathcal{A}_\circ$ is the $\mathcal{A}$ with $A^{ori}$ set to the zero matrix, and $J = R - 2R \circ A^{ori}$.

Given $J$, maximizing $\text{tr}(J^\top V)$ over $V \in \mathcal{A}_\circ^{2+3}$ is exactly the same problem as (10). Maximizing over $\mathcal{A}_\circ^{1+3}$ is a maximum weight $b$-matching problem ($b = \delta_l$) with a fully connected graph, and its exact solution can be found in $O(n^4)$ time [41]. Maximization over $\mathcal{A}_\circ^{1+2}$ can be solved by dynamic programming almost identical to Algorithm 1. Simply redefine $a_t(j)$ therein by the negative of the sum of the largest $j$ elements in $J_{t:}$ with $a_t(0) = 0$. It costs $O(n\delta_g\delta_l)$ in time, and $O(n\delta_g)$ in space.

The PO for $\mathcal{A}_\circ^{1+2+3}$ is NP-hard, but a $\frac{1}{2}$-approximate solution (in a relative sense) can be found by greedily adding edges to $V$ as shown in Algorithm 2. The time complexity is $O(n^2 \log n)$. The analysis, however, is rather involved based on $k$-extendible systems, and we summarize our bound and implied certificate in Theorem 1. By [42], CG with an $\alpha$-approximate PO converges to a solution whose objective value is at most $\frac{1}{\alpha}$ times of the true minimum (assume w.l.o.g. that it is positive).

**Theorem 1.** *(Proof relegated to Appendix A.1) Suppose Algorithm 2 returns $V^*$, and CG returns a solution $Z_t$. Then $\text{tr}(J^\top V^*) \geq \frac{1}{2} \max_{V \in \mathcal{A}_\circ^{1+2+3}} \text{tr}(J^\top V)$, and a certificate can be derived as*

$$\min_{A \in \mathcal{A}^{1+2+3}} F(A) \geq F^\circ(Z_t) + \text{tr}(R^\top Z_t) - 2\,\text{tr}(J^\top V^*) - \text{tr}(R^\top A^{ori}). \tag{13}$$

**Restoring tractability of PO by relaxing the convex hull.** Since $\text{co}\,\mathcal{A}^{1+2+3}$ only admits a $\frac{1}{2}$-approximate PO, the resulting certificate may be not tight. It is thus worthwhile to slightly relax the domain in exchange for the factor of 2. The most natural one is $\mathcal{C} := \text{co}\,\mathcal{A}^1 \cap \text{co}\,\mathcal{A}^2 \cap \mathcal{A}^3$ (recall $\mathcal{A}^3$ is already convex). PO on $\mathcal{C}$ simply optimizes a linear function over linear constraints $\{Z \in [0,1]^{n \times n} : Z^\top = Z, Z_{ii} = 1, \|Z - A^{ori}\|_1 \leq 2\delta_g, \|Z_{i:} - A_{i:}^{ori}\|_1 \leq \delta_l\}$. More details are given in Appendix A.2. In our experiment, the optimal $F^\circ$ value found is very close to what CG found for $\text{co}\,\mathcal{A}^{1+2+3}$.

## 4.2 Convexification of $F(A)$ for linear activation

We next convexify $F(A)$. To start with, let us consider the identity activation $\sigma$. Although $f_i$ is defined on the discrete domain, we can think of it as an extended function on a continuous domain:

$$h_i(z) := \begin{cases} f_i(z) & \text{if } z \in \mathcal{P}_i := \{z \in \{0,1\}^n : z_i = 1, \|z - A_{i:}^{ori}\|_1 \leq \delta_l\} \\ \infty & \text{otherwise} \end{cases}. \tag{14}$$

So the convex envelope of $h_i$ over $\text{co}\,\mathcal{P}_i$ can be written as

$$h_i^{**}(z) = \sup_r \left\{ r^\top z - \sup_{w \in \mathcal{P}_i} \left\{ r^\top w - f_i(w) \right\} \right\} = \max_r \min_{w \in \mathcal{P}_i} r^\top z - r^\top w + f_i(w). \tag{15}$$

This objective is concave in $r$, thus solvable by, *e.g.*, bundle method. Further, given $r$, the optimal $w$ can be found efficiently using the same technique as minimizing $F(A)$ over $\mathcal{A}^1$ (Appendix C.1). So it is straightforward to evaluate $h_i^{**}(z)$, and the convex envelope of $F(A)$ over $\text{co}\,\mathcal{A}^1 = \prod_i \text{co}\,\mathcal{P}_i$ is

$$F_1^\circ(Z) := \sum_i h_i^{**}(Z_{i:}), \qquad Z \in \text{co}\,\mathcal{A}^1. \tag{16}$$

Since all points in $\mathcal{P}_i$ must be an extreme point of $\text{co}\,\mathcal{P}_i$, $F_1^\circ(Z) = F(Z)$ for all $Z \in \mathcal{A}^1$. Although $F_1^\circ$ is derived from $\mathcal{A}^1$, it can be optimized over any subset of $\text{co}\,\mathcal{A}^1$, *e.g.*, $\text{co}\,\mathcal{A}^{1+2+3}$.

Further refinement is possible by considering all $Z_{i:}$ jointly to enforce the global budget: $H(Z) := \sum_i f_i(Z_{i:})$ if $Z \in \mathcal{A}^{1+2}$, and is $\infty$ otherwise. Then the convex envelope of $F(A)$ over co $\mathcal{A}^{1+2}$ is

$$F^\circ_{1+2}(Z) := H^{**}(Z) = \sup_R \left\{ \text{tr}(R^\top Z) - \sup_{W \in \mathcal{A}^{1+2}} \left\{ \text{tr}(R^\top W) - \sum_i f_i(W_{i:}) \right\} \right\}. \quad (17)$$

**Optimization recipe** (17) is again concave in $R$, and given $R$, the optimal $W$ can be found efficiently by Algorithm 1, providing the gradient in $R$. So given $Z$, the optimal $R$ can be found by bundle method. Then by Danskin's theorem, the gradient of $F^\circ_{1+2}(Z)$ in (17) is simply the optimal $R$, which can be fed to the polar operator (PO) for CG. Clearly, $F^\circ_{1+2}(Z) = F(Z)$ for all $Z \in \mathcal{A}^{1+2}$.

### 4.3 Convexification of $F(A)$ for ReLU activation

When $\sigma$ is ReLU, $f_i(A_{i:}) = (\hat{A}_{i:}\mathbf{1})^{-1}[\hat{A}_{i:}XW]_+ v$, where $v := \alpha_i(U_{:y} - U_{:c})$, and $[z]_+ := \max\{0, z\}$ is applied element-wise to a vector. Our first step is to adopt the technique in [12, 25], which approximates a ReLU unit $\hat{x} := [x]_+$ over $x \in [l, u]$ via the following intervals or values:

$$\hat{x} \in [[x]_+, u(x - l)/(u - l)] \text{ if } l \cdot u < 0, \qquad \hat{x} = x \text{ if } l \geq 0, \qquad \hat{x} = 0 \text{ if } u \leq 0. \quad (18)$$

Both $l$ and $u$ of the $j$-th entry of $\hat{A}_{i:}XW$ can be estimated under $\left\| A_{i:} - A_{i:}^{ori} \right\|_1 \leq \delta_l$ (Appendix C.2). Since $f_i(A_{i:})$ is to be minimized, minimizing $v_j \hat{x}_{ij}$ under (18) with $x = x_{ij} = \hat{A}_{i:}XW_{:j}$ yields

$$\min_{\hat{x}_{ij} \text{ meets (18)}} v_j \hat{x}_{ij} = \begin{cases} v_j \cdot \sigma^-_{ij}(x_{ij}) := v_j u_{ij}(\hat{A}_{i:}XW_{:j} - l_{ij})/(u_{ij} - l_{ij}) & \text{if } j \in \mathcal{N}_i \\ v_j \cdot \sigma^+_{ij}(x_{ij}) := v_j[\hat{A}_{i:}XW_{:j}]_+ & \text{otherwise} \end{cases}, \quad (19)$$

$$\text{where} \quad \mathcal{N}_i := \{j : v_j < 0 \wedge l_{ij} \cdot u_{ij} < 0\}. \quad (20)$$

Letting $e_i$ be the canonical vector for coordinate $i$, we can lower bound $f_i(A_{i:})$ by $g_i(A_{i:})$, where

$$g_i(z) := (\mathbf{1}^\top z + 1)^{-1} \hat{g}_i(z), \quad z \in \mathcal{P}_i \quad (21)$$

$$\text{where} \quad \hat{g}_i(z) := \sum_{j \notin \mathcal{N}_i} v_j \sigma^+_{ij}((z + e_i)^\top XW_{:j}) + \sum_{j \in \mathcal{N}_i} v_j \sigma^-_{ij}((z + e_i)^\top XW_{:j}). \quad (22)$$

To finally convexify it, note that $\sigma^-_{ij}$ is affine, hence can be handled with ease. $\sigma^+_{ij}$ is in fact the original ReLU, which yields the greater of 0 and an affine function. So $\hat{g}_i$ is convex. We now extend the convex envelope technique to $g_i$. First its form in (21) allows it to be trivially extended to co $\mathcal{P}_i$:

$$h_i(z) = g_i(z) = (\mathbf{1}^\top z + 1)^{-1} \hat{g}_i(z) \quad \text{if } z \in \text{co } \mathcal{P}_i, \qquad \text{and } h_i(z) = \infty \quad \text{otherwise}. \quad (23)$$

Then the convex envelope of $h_i$ over co $\mathcal{P}_i$ can be derived as

$$h_i^{**}(z) = \sup_r \left\{ r^\top z - \sup_{w \in \text{co } \mathcal{P}_i} \left\{ r^\top w - (\mathbf{1}^\top w + 1)^{-1} \hat{g}_i(w) \right\} \right\} \quad (24)$$

$$= \max_r \min_{\alpha \in [0,n]} \min_{w : w \in \text{co } \mathcal{P}_i, \, \mathbf{1}^\top w = \alpha} \{ r^\top z - r^\top w + (\alpha + 1)^{-1} \hat{g}_i(w) \}. \quad (25)$$

Given $r$ and $\alpha$, the inner-most optimization over $w$ is convex because $\hat{g}_i$ is convex. The optimization over $\alpha$ is not convex, but it is one dimensional, hence can be solved globally with $O(1/\epsilon)$ complexity, e.g., by enumerating over an $\epsilon$-grid. The formula of the derivative in $\alpha$ is relegated to Appendix C.3. We solved $r$ by bundle method. The final convexified objective is $F^\circ_{1+2}(Z) := \sum_i h_i^{**}(Z_{i:})$.

## 5 Experimental Results

Our experiment aims to evaluate the certificate of robustness for GCNs in graph classification, with an emphasis on its tightness. This is facilitated by the ADMM that provides an upper bound under topological attacks. We will also apply the certificate to the models trained with a robust regularizer. All code and data are available at https://github.com/RobustGraph/RoboGraph.

**Datasets** We tested on four public datasets that are commonly used in graph classification: Enzymes, NCI1, MUTAG, and PROTEINS [43]. Their properties are summarized in Appendix E. We used 30% of the whole dataset for training, 20% for validation, and 50% for testing. The test set has a higher fraction because our focus is on attacking the model on the test set. Due to space limit, we only report the result on Enzymes, relegating to Appendix E the result of the other two datasets. Enzymes has 600 graphs, $C = 6$ classes, $d = 21$ node features. The median number of node over all graphs is 32, while that of edge is 120.

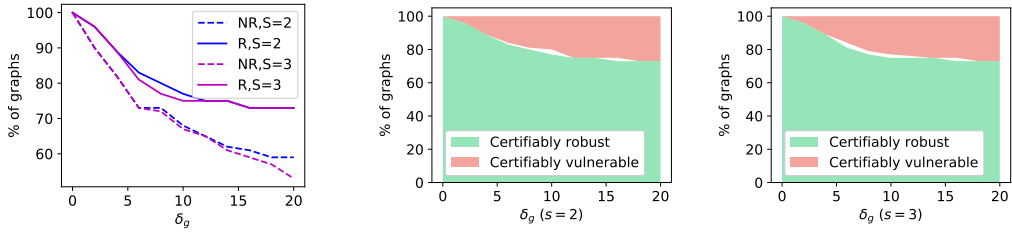

(a) Fraction of graphs certified robust with $s \in \{2, 3\}$, under robust training (**R**) and non-robust training (**NR**)

(b) Fraction of graphs that are certified as robust (lower green area) and vulnerable (upper red area, percentage $= 100 - y$-axis). Left: $s = 2$, right: $s = 3$ for attack. Both are under robust training.

Figure 1: Certificates under **linear** activation on Enzymes. $s$ is the local attack strength at *testing*.

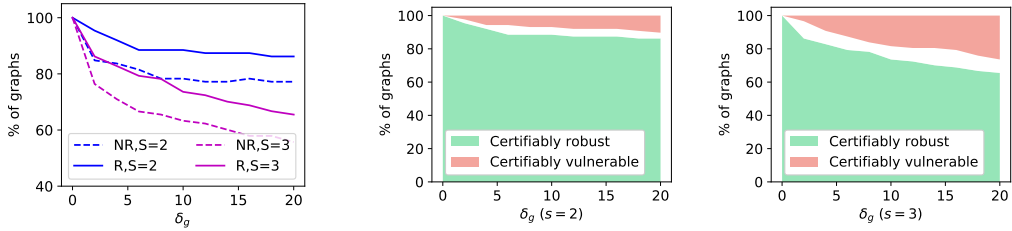

(a) Fraction of graphs certified robust

(b) Fraction of graphs that are certified as robust and vulnerable

Figure 2: Certificates under **ReLU** activation on Enzymes, with other settings identical to Figure 1

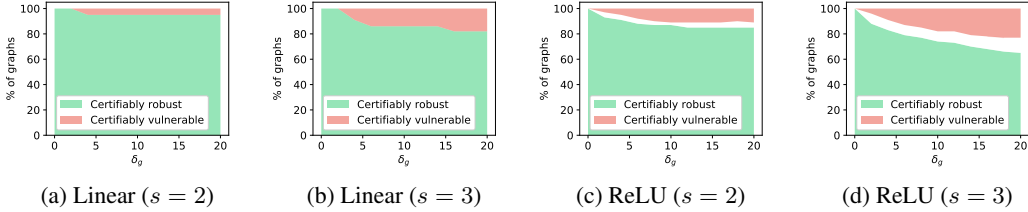

(a) Linear ($s = 2$)  (b) Linear ($s = 3$)  (c) ReLU ($s = 2$)  (d) ReLU ($s = 3$)

Figure 3: Certificate for Enzymes with 80% for training. Settings are identical to Figure 1 and 2.

**Training algorithms and settings**    We used a single hidden layer and the latent dimensionality was set to 32. Average pooling was also deployed. In [12], the local budget was set to 1% of the total local degree of freedom (*i.e.*, number of node features). This is clearly inapplicable to our setting because most graphs do not even have 100 nodes; indeed, the median #node is 32. Following [19], we varied the local budget for each node $v$ in a graph $G$ by $\delta_l(v) = \max(0, d_v - \max(\{d_v : v \in G\}) + s)$, where $d_v$ is the degree of $v$ in the original graph, $\max(\{d_v : v \in G\})$ is the max of the node degrees in $G$, and $s$ called the *local attack strength* that is varied from 1, 2, etc. A lower value of $s$ results in a more restrictive budget. The underlying rationale is to endow a larger local budget for higher degree nodes, because they often require more perturbations to make a difference [11, 12]. Empirically we observed that when $s = 3$, about 60% of the nodes in each graph get a positive budget thanks to ties in $d_v$, and that rate goes to nearly 100% when $s = 4$.

As shown by [12, 19], a hinge loss added to (1) can significantly improve the robustness without degrading the accuracy: $\lambda \sum_{c \neq y} \max\{0, 1 + \max_{A \in \mathcal{A}}(z_c(A) - z_y(A))\}$, adapted from node classification to graph classification. So in practice, it is almost always advisable to use it, computation permitting. In our experiment, we set $\lambda = 0.5$, $s^{tr} = 3$, and $\delta_g^{tr} = \infty$, because they produced the highest robustness while not hurting the test accuracy. Note that the value of $s$ and $\delta_g$ for test data can be different from $s^{tr}$ and $\delta_g^{tr}$ (*i.e.*, different $\mathcal{A}$). Our work claims no novelty in robust training, and it was leveraged to demonstrate the effectiveness of our certificate in a more realistic setting.

**Performance of the certificates**    Figure 1a demonstrates the percentage of graphs that are certified robust by the dualization method, where the activation function is linear. Figure 2a shows a similar result, but for ReLU activation using the convex envelope approach from Section 4.3, optimized

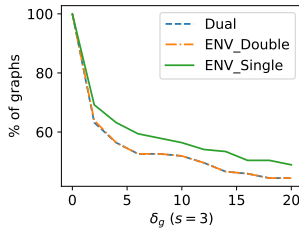

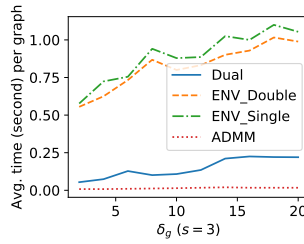

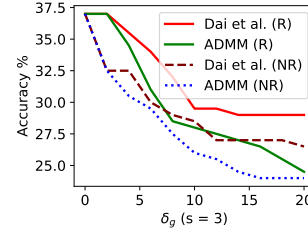

Figure 4: Comparison of convex certificates on ReLU

Figure 5: Computational time for certificates/attacks on ReLU

Figure 6: Test accuracy under various attacks

over $\text{co }\mathcal{A}^1 \cap \text{co }\mathcal{A}^2 \cap \mathcal{A}^3$. The global budget $\delta_g$ is varied from 1 to 20. It is clear that robust training significantly increases the fraction of graphs that are certifiably robust. Naturally, $s = 2$ allows more graphs to be certified robust. Since node features are not subject to attack, the certified percentage does not drop to 0 even for large $\delta_g$, akin to Figure 4a of [19].

**Tightness of the certificates**    To better examine the tightness of our certificates, Figure 1b plots the percentage of graphs that are certified as robust (lower green area, same as the 'R' curves Figure 1a) and vulnerable (by ADMM, upper red area). It is for linear activation under robust training, and that for ReLU is in Figure 2b. Clearly the gap is almost zero for linear activation, while that for ReLU is also quite small, below 20% which is similar to [12]. So the gap stems from the approximation in ReLU, and any improvement in this respect can benefit a number of certificate algorithms like ours.

In addition, we compared three convex certificates for ReLU: ENV_Single (used above) is the convex envelop based on the single linear approximation (for $\mathcal{N}_i$) as in (19). We also adopted the double linear approximation (for $\bar{\mathcal{N}}_i$) from [26], which enables both dualization (Dual) and convex envelope (ENV_Double) methods. Figure 4 shows ENV_Single yields clearly superior (higher) certificate.

**Performance on larger training sets**    We additionally experimented on Enzyme with 80%, 10%, 10% for training, validation and testing, respectively. The small size of test data led to marked variations in the gap plot, and it is unclear how to "average" them. So we plotted a typical result in Figure 3, showing the fraction of certifiably robust/vulnerable for both linear and ReLU activations. Compared with Figure 1b and 2b where 30% graphs were used for training, the tightness here appears similar, or slightly better under the linear activation.

**Computational efficiency**    The two convex envelope based methods are more costly (Figure 5). Since single linear approximation keeps the nonsmooth $[\cdot]_+$ function in (19) for $\bar{\mathcal{N}}_i$, it is slightly more expensive than double linear approximation.

To test the scalability to graphs with a larger number of nodes and edges, we examined another dataset DD [44], where the median number of node and edge per graph is 284 and 716, respectively. We set $\delta_g = 20$ and $s = 3$. For the certificates with linear activation, Enzyme (median #node = 32, median #edge = 120) takes 0.37 seconds per graph on average, while DD takes 7.3 seconds. For the certificates with ReLU activation, Enzyme takes 1 second per graph on average, while DD takes 28 seconds. This is consistent with the fact that the computational cost depends quadratically on the number of nodes (in practice, a bit lower than that due to implementation details).

Different datasets have different number of classes. To facilitate comparison, the reported time cost is for each class. The number of edges in the original graph does not affect the computational cost much, because the attacker can both delete **and** add edges.

**Comparison with other structural attacks**    While not being our key focus, we also compared ADMM with Dai et al. [10], which provides an effective topological attack to a flexible range of applications including graph classification. [11, 12, 15, 19] are not applicable to structural attacks on GCNs for graph classification. As can be seen from Figure 6, robust training (R) mitigates the drop of accuracy compared with non-robust training (NR), and ADMM finds more effective attacks under both robust and non-robust training.

**Conclusion**    In this paper we proposed the first robustness certificate for graph classification using GCNs under structural attacks. Its tightness and efficiency, along with a new ADMM-based attack, are demonstrated empirically. We will extend the framework to distributionally robust optimization.

## Acknowledgements

We thank the reviewers for their constructive comments. This work is supported by Google Cloud and NSF grant RI:1910146.

## Broader Impact

Graph convolutional networks (GCNs) have been shown effective in a number of applications such as social networks, biological graphs, citation networks, and etc. Despite its recent success, its vulnerability to adversarial attacks has also been revealed and attacks on both node feature and graph structure have been proposed. Direct extension of defense algorithms from image classification domain meets with immediate obstacles because computing the adversarial network is a highly involved discrete optimization problem, costing a substantial amount of computations.

This paper proposed the first algorithm that provides tight lower and upper bounds for the margin of **graph classification** under both global and local budget constraints, allowing a certificate of robustness to be computed efficiently in practice, and proved in theory. It can be readily applied to a number of high-impact domains in real-world problems, including cross-lingual knowledge graph alignment [45], fraud detection [46], cancer classification based on multi-modal fMRI images [47], chemical and biological interface prediction [48], categorization of scientific papers [1], etc.

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
