[Supplementary Material]

# Supplementary Material

## A Proofs

### A.1 Proof of Theorem 1.

The proof for the first part follows from [49], which showed that if the constraint set is a $k$-extendible system, then the greedy algorithm can find a $\frac{1}{k}$-approximate solution. So it suffices to check that $\mathcal{A}_{\circ}^{1+2+3}$ is 2-extendible.

Let $T$ be a finite set and $\mathcal{L}$ be a collection of subsets of $T$. Then $(T, \mathcal{L})$ is 2-extendible if

- For all $C \subseteq D$, if $D \in \mathcal{L}$ then $C \in \mathcal{L}$;
- Suppose $C \subseteq D \in \mathcal{L}$, and $x$ be such that $x \notin C$ and $C \cup \{x\} \in \mathcal{L}$. Then there exists $Y \subseteq D \backslash C$ such that $|Y| \leq k$ and $D \backslash Y \cup \{x\} \in \mathcal{L}$.

Consider the obvious bijection between $V \in \mathcal{A}_{\circ}^{1+2+3}$ and a graph of $n$ nodes where two different nodes $i$ and $j$ are connected by an undirected edge if, and only if, $(X_{ji} =) X_{ij} = 1$. So let $T$ be the set of all possible (non-self) undirected edges, and $\mathcal{L}$ be the set of undirected graphs with at most $\delta_g$ edges and each node has degree at most $\delta_l$. Clearly such $(T, \mathcal{L})$ satisfies the first condition. To check the second condition, let $x$ be an edge $(i, j)$. If $x \in D$, then the condition holds trivially with $Y = \emptyset$. Otherwise, $x \notin D$, hence $x \notin C$. Note that the degree of $i$ and $j$ in the graph $C$ must be strictly less than $\delta_l$, because otherwise $C \cup \{x\}$ would not be in $\mathcal{L}$ (*i.e.*, not valid). If the degree of $i$ in $D$ is $\delta_l$, then we can find an edge $e_1 \in D \backslash C$ that is incident to $i$, and add $e_1$ to $Y$. Similarly if the degree of $j$ in $D$ is $\delta_l$, then we can find an edge $e_2 \in D \backslash C$ that is incident to $j$, and add $e_2$ to $Y$. Clearly $|Y| \leq 2$, $Y \subseteq D \backslash C$, and $D \backslash Y \cup \{x\} \in \mathcal{L}$.

To prove the second part, note by convexity, $F(A) = F^{\circ}(A) \geq F^{\circ}(Z_t) + \operatorname{tr}((A - Z_t)^{\top} \nabla F^{\circ}(Z_t))$. Therefore

$$\min_{A \in \mathcal{A}} F(A) \geq F^{\circ}(Z_t) + \operatorname{tr}(R^{\top} Z_t) - \max_{A \in \mathcal{A}} \operatorname{tr}(R^{\top} A), \quad \text{where} \quad R = -\nabla F^{\circ}(Z_t). \tag{26}$$

If the PO can be solved exactly, then the right-hand side can be evaluated exactly, leading to a slightly tighter certificate than $F^{\circ}(Z_t)$. However, if the PO is not tractable, then let $V^*$ be the result returned by the greedy algorithm, and it follows from Theorem 1 that

$$\max_{A \in \mathcal{A}^{1+2+3}} \operatorname{tr}(R^{\top} A) = \max_{V \in \mathcal{A}_{\circ}^{1+2+3}} \operatorname{tr}(R^{\top}((-2A^{ori} + E) \circ V + A^{ori})) \tag{27}$$

$$\leq 2 \operatorname{tr}(J^{\top} V^*) + \operatorname{tr}(R^{\top} A^{ori}), \quad \text{where} \quad J = R \circ (-2A^{ori} + E). \tag{28}$$

Plugging it into (26) and we get a feasible certificate, though at a price of 2.

### A.2 Representing $\operatorname{co} \mathcal{A}^1 \cap \operatorname{co} \mathcal{A}^2 \cap \mathcal{A}^3$ by linear inequalities.

We first show that

$$\operatorname{co} \mathcal{D} = \{z \in [0, 1]^n : \|z - \alpha\|_1 \leq \delta_l\}, \quad \text{where} \quad \mathcal{D} := \{z \in \{0, 1\}^n : \|z - \alpha\|_1 \leq \delta_l\}. \tag{29}$$

Clearly, the right-hand side subsumes $\mathcal{D}$ and is convex. Therefore it subsumes $\operatorname{co} \mathcal{D}$. To show the converse direction, it suffices to show that for any $\gamma \in \mathbb{R}^n$, $\max_{z \in z \in [0,1]^n : \|z - \alpha\|_1 \leq \delta_l} \gamma^{\top} z$ can be attained at an integral solution (hence in $\mathcal{D}$). To this end, without loss of generality, suppose $\alpha_1 = \ldots = \alpha_m = 0$ and $\alpha_{m+1} = \ldots = \alpha_n = 1$, where $m \in \{0, 1, \ldots, n\}$. Then $\|z - \alpha\|_1 \leq \delta_l$ can be written as $\sum_{i=1}^m z_i - \sum_{i=m+1}^n z_i \leq \delta_l - n + m$. So this linear inequality, along with $z_i \leq 1$, can be written as $Pz \leq (1, \ldots, 1, \delta_l - n + m)^{\top}$, where

$$P := \begin{pmatrix} 1 & & & & & \\ & 1 & & & & \\ & & 1 & & & \\ & & & 1 & & \\ & & & & 1 & \\ & & & & & 1 \\ 1 & \ldots & 1 & -1 & \ldots & -1 \end{pmatrix} \in \mathbb{R}^{(n+1) \times n}. \tag{30}$$

Here the last row has $m$ ones, followed by $n - m$ copies of $-1$. Now we can partition the rows of $P$ into two groups $R_1$ and $R_2$, where $R_1$ consists of the first $m$ rows, and $R_2$ consists of the remaining $n - m + 1$ rows. Now obviously each column contains at most two nonzero entries. For the first $m$ columns, the two nonzero entries have the same sign, with one belonging to $R_1$ and the other to $R_2$. For the last $n - m + 1$ columns, both the 1 and $-1$ belong to $R_2$. So $P$ is totally unimodular [50]. Finally noting that $\delta_l - n + m$ is integral, there must be an optimal solution that is integral.

## B  ADMM Properties

Although ADMM is not guaranteed to find the global optimum, the analogy with convex ADMM suggests that it may well minimize $F(A)$ approximately. This is verified in our experiment, and it trivially provides an upper bound for $F(A)$ under $A \in \mathcal{A}^{1+2+3}$.

In practice, we can add additional constraints to $B$ in the splitting formula (7), hoping that the overlap of constraints with $A$ can help accelerate convergence. For example, if we replace $\delta(B \in \mathcal{A}^3)$ by $\delta(B \in \mathcal{A}^{2+3})$, then the update of $B_t$ in (10) still admits a closed form because the objective is linear in $B$ (note $x^2 = x$ for $x \in \{0, 1\}$).

In particular, since $B$ must be symmetric, the linearity of the objective allows it to be written as $\mathrm{tr}(Q^\top B)$ for some symmetric matrix $Q$. So we only need to sort the entries $\{Q_{ij} : i < j\}$ in an ascending order. If there are at least $\delta_g$ negative numbers in it, then take the first $\delta_g$ indices and set their corresponding entries in $B$ to 1, with the rest set to 0. If there are less than $\delta_g$ negative numbers, then find their indices and set the corresponding entries in $B$ to 1. Finally mirror the 1's to the lower triangle Overall, the computational cost is $O(n^2 \log n)$.

Similarly, to optimize $F(A)$ over $\mathcal{A}^{1+3}$ and $\mathcal{A}^{2+3}$, simply use $\delta(A \in \mathcal{A}^1) + \delta(B \in \mathcal{A}^{1+3})$ and $\delta(A \in \mathcal{A}^2) + \delta(B \in \mathcal{A}^{2+3})$, respectively. In both cases, the optimization in $A$ is over simple constraints, while that over $B$ can be done as above.

To further improve the solution, we also applied local adjustment by looping over:

**Pruning**: if removing an edge can improve $F$, then pick one that improves $F$ best.

**Adding**: if the local/global budget allows, then add an edge that best improves $F$.

**Replacing**: If removing an edge and adding a new one improves $F$, then find the substitution that best improves $F$ while respecting the local budgets

The process can be terminated if no more change is made to the graph in the loop.

## C  Algorithmic Details

### C.1  Optimizing $f_i(A_{i:})$ over $\mathcal{P}_i := \{z \in \{0,1\}^n : z_i = 1, \left\| z - A_{i:}^{ori} \right\|_1 = j\}$

Recall from (3) that $f_i(A_{i:}) = \alpha_i \sigma \left( (\hat{A}_{i:} \mathbf{1})^{-1} \hat{A}_{i:} XW \right) (U_{:y} - U_{:c})$. When $\sigma$ is identity, we can write it as $f_i(A_{i:}) = (\hat{A}_{i:} \mathbf{1})^{-1} \hat{A}_{i:} \pi$ for some vector $\pi$. Denote $a = (A_{i:}^{ori})^\top$ and let $v_k$ encode whether $A_{ik}$ changes upon $A_{ik}^{ori}$ (1 for true and 0 for false). Then $A_{i:}^\top = v + a - 2a \circ v$. Noting that $\mathbf{1}^\top v = j$, the optimization can now be written as

$$\min_v f(v) = \frac{\beta^\top v + c_1}{a^\top v + c_2}, \quad s.t. \quad v \in \{0,1\}^n, \ \mathbf{1}^\top v = j, \ v_i = 0, \tag{31}$$

for some vector $\beta$ and scalar $c_1, c_2$. Now we only need to enumerate all possible values of $s := a^\top v$ from 0 to $\min(j, \mathbf{1}^\top a)$ in the denominator. Let $I_+ := \{k \in [n] : a_k = 1\}$ and $I_- = [n] \backslash (\{i\} \cup I_+)$. Then it is trivial to optimize the numerator under $a^\top v = s$ by computing

$$\min_{v_k : k \in I_+} \sum_{k \in I_+} \beta_k v_k, \quad s.t. \quad v_k \in \{0,1\}, \ \sum_{k \in I_+} v_k = s, \tag{32}$$

$$\min_{v_k : k \in I_-} \sum_{k \in I_-} \beta_k v_k, \quad s.t. \quad v_k \in \{0,1\}, \ \sum_{k \in I_-} v_k = j - s. \tag{33}$$

Both can be computed by sorting $\{\beta_k : k \in I_+\}$ and $\{\beta_k : k \in I_-\}$, and this sorting only needs to be done once (and be used for all values of $s$). So the overall complexity of optimizing $f_i(A_{i:})$ over $\mathcal{P}_i$ is $O(n \log n)$.

## C.2  Lower and upper bound for ReLU approximation

Both $l$ and $u$ of the $j$-th entry of $\hat{A}_{i:}XW$ can be easily estimated under $\left\|A_{i:} - A_{i:}^{ori}\right\|_1 \le \delta_l$. Let $V = |A_{i:} - A_{i:}^{ori}|$, so that $V_j = 0$ if $A_{i:}$ makes no change to $A_{ij}^{ori}$, and $V_j = 1$ otherwise (adding or removing an edge). Then

$$l = \min_{\left\|A_{i:} - A_{i:}^{ori}\right\|_1 \le \delta_l} A_{i:}(XW)_{:j} + (XW)_{i,j} \tag{34}$$

$$= \min_{\mathbf{1}^\top V \le \delta_l} \frac{(2A_{i:}^{ori} - 1) \circ (-2V^\top + 1) + 1}{2}(XW)_{:j} + (XW)_{i,j} \tag{35}$$

$$= \min_{\mathbf{1}^\top V \le \delta_l} [(1 - 2A_{i:}^{ori}) \circ (XW)_{:j}^\top]V + A_{i:}^{ori}(XW)_{:j} + (XW)_{i,j} \tag{36}$$

Now the detailed algorithm can be derived and is presented in Algorithm 3.

---

**Algorithm 3:** $l$ and $u$ of the $j$-th entry of $\hat{A}_{i:}XW$

---

**1** Initialize $V^l = V^u = \mathbf{0}$ and set $J = (1 - 2A_{i:}^{ori}) \circ (XW)_{:j}^\top$.
**2** Sort the indices $\mathcal{I} := \{1, \ldots, n\}$ in an ascending order of $J$.
**3** Let $k = 1$
**4** **while** $J[\mathcal{I}[k]] < 0$ **and** $\mathbf{1}^\top V^l < \delta_l$ **and** $\mathbf{1}^\top V^l < \delta_g$ **do**
**5**   $\quad$ If $\mathcal{I}[k] \ne i$, set $V^l[\mathcal{I}[k]] = 1$
**6**   $\quad$ $k = k + 1$
**7** **Return** $V^l$
**8** $l = [(1 - 2A_{i:}^{ori}) \circ (XW)_{:j}^\top]V^l + A_{i:}^{ori}(XW)_{:j} + (XW)_{i,j}$
**9** Let $k = n$
**10** **while** $J[\mathcal{I}[k]] > 0$ **and** $\mathbf{1}^\top V^u < \delta_l$ **and** $\mathbf{1}^\top V^u < \delta_g$ **do**
**11**   $\quad$ If $\mathcal{I}[k] \ne i$, set $V^u[\mathcal{I}[k]] = 1$
**12**   $\quad$ $k = k - 1$
**13** **Return** $V^u$
**14** $u = [(1 - 2A_{i:}^{ori}) \circ (XW)_{:j}^\top]V^u + A_{i:}^{ori}(XW)_{:j} + (XW)_{i,j}$

---

## C.3  Derivative of (25) in $\alpha$.

Note we can change variable by $\theta = |w - A_{i:}^{ori}|$, so that $\theta_j = 0$ if $w$ makes no change to $A_{ij}^{ori}$, and $\theta_j = 1$ otherwise (adding or removing an edge). Then $w = \frac{1}{2}[(2A_{i:}^{ori} - 1) \circ (-2\theta + 1) + 1]$. So $\mathbf{1}^\top w = \alpha$ can be translated into a linear constraint on $\theta$, which we denote as $\beta^\top \theta = \eta_\alpha$. Now $w \in \text{co}\,\mathcal{P}_i$ is equivalent to $\theta_j \in [0, 1]$ and $\mathbf{1}^\top \theta \in [0, \delta_l]$. Write out the Lagrangian for the minimization over $w$:

$$J(\alpha) := \min_{\theta : \mathbf{1}^\top \theta \le \delta_l, \beta^\top \theta = \eta_\alpha, \theta_j \in [0,1]} (\alpha + 1)^{-1} \kappa(\theta) - \gamma^\top \theta \tag{37}$$

$$= \min_\theta \max_{\lambda \ge 0, \mu, \rho_j \ge 0, \xi_j \ge 0} (\alpha + 1)^{-1} \kappa(\theta) - \gamma^\top \theta + \lambda(\mathbf{1}^\top \theta - \delta_l) + \mu(\beta^\top \theta - \eta_\alpha) \tag{38}$$

$$+ \sum_j \rho_j(\theta_j - 1) - \sum_j \xi_j \theta_j. \tag{39}$$

Taking partial derivative with respect to $\theta_j$, we have

$$(\alpha + 1)^{-1} \nabla_j \kappa(\theta) - \gamma_j + \lambda + \mu \beta_j + \rho_j - \xi_j = 0. \tag{40}$$

If $\theta_j \in (0, 1)$, then $\rho_j = \xi_j = 0$, and

$$(\alpha + 1)^{-1} \nabla_j \kappa(\theta) - \gamma_j + \lambda + \mu \beta_j = 0. \tag{41}$$

So as long as there are two indices $j$ which satisfy $\theta_j \in (0, 1)$, we can solve for $(\lambda, \mu)$. If $\mathbf{1}^\top \theta < \delta_l$, we can further simplify by $\lambda = 0$. In practice, we can collect all such $j$ and find the least square solution of $(\lambda, \mu)$. With that, we can compute $J'(\alpha) = -\kappa(\theta)(\alpha + 1)^{-2} - \mu \frac{\partial}{\partial \alpha} \eta_\alpha$.

## D  Extension to Multiple Hidden Layers

As noted by [1], GCNs do not benefit from more than two hidden layers. For completeness, we sketch here how our approach can be extended to two hidden layers. In this case

$$F(A) = \sum_{i=1}^n \alpha_i \underbrace{L_{i:}\sigma(LXW)(U_{:y} - U_{:c})}_{=:f_i(A_{i:})}, \quad \text{where} \quad L := \hat{D}^{-1}\hat{A}. \tag{42}$$

$F$ is quadratic in $L$ when $\sigma$ is the linear activation. When $\sigma$ is ReLU, we can use the double linear approximation as in [26], and it will again make $F$ quadratic in $L$. As a result, in both cases, the second-order Taylor expansion will be exact

$$F(L) = F(L^{ori}) + \text{tr}((L - L^{ori})^\top \nabla F(L^{ori})) + \tfrac{1}{2}\text{vec}(L - L^{ori})^\top \cdot H \cdot \text{vec}(L - L^{ori}), \tag{43}$$

where we vectorized $L^{ori}$ so that $H := \nabla^2 F(\text{vec}(L))$ is a $n^2$-by-$n^2$ matrix, which is in fact independent of $L$ because $F$ is quadratic in $L$. Letting

$$\sigma_F := \max\{\text{vec}(V)^\top \cdot H \cdot \text{vec}(V) : V \in \mathbb{R}^{n \times n}, \|V\|_F \leq 1\}, \tag{44}$$

then

$$F(L) \geq F(L^{ori}) + \text{tr}((L - L^{ori})^\top \nabla F(L^{ori})) - \tfrac{\sigma_F}{2}\left\|L - L^{ori}\right\|_F^2. \tag{45}$$

To minimize the right-hand side of (45), notice that all terms linear in $L$ can be dealt with in the same way as in one-hidden-layer GCNs. The only new term is $\|L\|_F^2$, which is equal to $\sum_i (A_{i:}^\top \mathbf{1})^{-1}$. Since the dynamic programming in Algorithm 1 is based on $A_{i:}^\top \mathbf{1}$ (see Appendix C), it can be easily extended to handle the extra terms arising from $\|L\|^2$.

The bound in (45) can be tightened in two major ways:

**1)** The norm can be refined. For example, instead of Frobenious norm, we can adopt $\left\|L - L^{ori}\right\|_{2,\infty} := \max_i \left\|L_{i:} - L_{i:}^{ori}\right\|_2$, which is equal to $\max_i (A_{i:}^\top \mathbf{1})^{-1/2}$. The major benefit is that this norm cannot be greater than 1, so squaring it in (45) is indeed a reduction. On the flip side, this new norm will complicate the computation of $\sigma_F$, so we propose the following semi-definite relaxation which offers a $\log n$-approximate solution.

Denote $X = \text{vec}(V)\text{vec}(V)^\top \in \mathbb{R}^{n^2 \times n^2}$. Then $\text{vec}(V)^\top \cdot H \cdot \text{vec}(V) = \text{tr}(HX)$ and the constraint that $\|V\|_{2,\infty} \leq 1$ implies

$$\sum_{i=1}^n X_{(i-1)n+t,(i-1)n+t} \leq 1, \quad \forall t \in [n]. \tag{46}$$

So $\sigma_F$ is upper bounded by an SDP relaxation on $X$:

$$\max_X \text{tr}(HX), \quad s.t. \quad X \succeq \mathbf{0}, \sum_{i=1}^n X_{(i-1)n+t,(i-1)n+t} \leq 1, \quad \forall t \in [n]. \tag{47}$$

Then by

A. Nemirovski, C. Roos, and T. Terlaky. On maximization of quadratic form over intersection of ellipsoids with common center. Math. Program. Ser. A, 86:463–473, 1999.

the optimal SDP objective is at most $2 \log(2n)$ times of the true value of $\sigma_F$ under $\|\cdot\|_{2,\infty}$.

**2)** The domain of $v$ considered in (44) only needs to cover the subset of the unit ball (under the above refined norm) that is attainable by $L - L^{ori}$ for some $A \in \mathcal{A}$. It can be much smaller than the value of $\sigma_F$ computed from the unit ball.

# E  Experiment

Here we provide detailed experiments for all datasets. Although part of the results for Enzymes have been shown in Section 5, we will further provide the results when the attack strength is $s = 4$ at testing. The observations and conclusions from all datasets are similar to what is presented in Section 5. The properties of all datasets are summarized in Table 1.

Table 1: Datasets for used in experiment.

| dataset | #graphs | #labels | # node features | median #node | median #edge |
|---------|---------|---------|-----------------|--------------|--------------|
| Enzymes | 600 | 6 | 21 | 32 | 120 |
| NCI1 | 4110 | 2 | 37 | 27 | 58 |
| PROTEINS | 1113 | 2 | 4 | 26 | 98 |
| MUTAG | 188 | 2 | 7 | 17 | 38 |

## E.1  Comparing activation and pooling functions

We first show in Table 2 that the performance of (linear, ReLU) activation, in conjunction with various pooling methods, can be quite mixed. No combination is uniformly the best. In particular, we considered average pooling (avg), max pooling (max), and attention pooling with

- att_node: the attention weights $\alpha$ were trained as functions of node features only;
- att_topo: the attention weights $\alpha$ were trained also using the graph Laplacian [51].

| dataset | activation | pooling | | | |
|---------|------------|---------|---------|----------|----------|
| | | avg | max | att_topo | att_node |
| Enzymes | ReLU | **31.6 ± .5** | 29.8 ± .4 | 29.5 ± 1.1 | 19.9 ± 1.3 |
| | Linear | 29.1 ± 1.7 | **30.3 ± .0** | 30.1 ± .8 | 21.3 ± 4.4 |
| NCI1 | ReLU | 65.0 ± 0.3 | 62.5 ± .0 | **67.6 ± .2** | 63.0 ± .1 |
| | Linear | 58.3 ± .0 | 62.5 ± .3 | 61.4 ± .0 | **63.2 ± .2** |
| PROTEINS | ReLU | **67.4 ± 1.2** | 66.9 ± 1.6 | 66.0 ± .0 | 64.9 ± .0 |
| | Linear | 64.1 ± 3.3 | **69.5 ± .4** | 65.5 ± .1 | 62.9 ± 2.3 |
| MUTAG | ReLU | 68.8 ± 1.5 | 66.1 ± .8 | 70.2 ± .4 | **73.3 ± 3.5** |
| | Linear | 65.3 ± .0 | 65.7 ± .8 | 69.4 ± 2.0 | **74.1 ± 2.0** |

Table 2: Comparison of graph classification accuracy under various activations and pooling functions. The best result of each row is marked in boldface. 30% data were used for training, 20% for validation, and 50% for testing. There is one hidden layer with $d' = 64$ hidden nodes. All settings were run for 10 times to obtain mean and standard deviation.

As a result, it is meaningful and useful to study the robustness certificate and attack for all combinations of activation and pooling.

## E.2 More results on Enzymes

(a) Linear activation

(b) ReLU activation

Figure 7: Fraction of graphs certified robust with $s \in \{2, 3, 4\}$, under robust training (**R**) and non-robust training (**NR**). Dataset: Enzymes.

(a) $s = 2$

(b) $s = 3$

(c) $s = 4$

Figure 8: Fraction of graphs that are certified as robust (lower green area) and vulnerable (upper red area, percentage $= 100 - y$-axis. **Linear activation.** All are under robust training. Dataset: Enzymes.

(a) $s = 2$

(b) $s = 3$

(c) $s = 4$

Figure 9: Same as Figure 8, but using **ReLU activation.**

(a) $s = 3$

(b) $s = 6$

Figure 10: Test accuracy under various attacks, and robust training (**R**) or non-robust training (**NR**). ReLU activation. Dataset: Enzymes.

(a) Linear activation        (b) ReLU activation

Figure 11: Fraction of graphs certified robust with $s \in \{2, 3, 4\}$, under robust training (**R**) and non-robust training (**NR**). Dataset: NCl1.

(a) $s = 2$        (b) $s = 3$        (c) $s = 4$

Figure 12: Fraction of graphs that are certified as robust (lower green area) and vulnerable (upper red area, percentage $= 100 - y$-axis. **Linear activation.** All are under robust training. Dataset: NCl1.

(a) $s = 2$        (b) $s = 3$        (c) $s = 4$

Figure 13: Same as Figure 12, but using **ReLU activation.**

(a) $s = 3$        (b) $s = 6$

Figure 14: Test accuracy under various attacks, and robust training (**R**) or non-robust training (**NR**). ReLU activation. Dataset: NCl1.

In general, the gap for linear activation should be smaller than that for ReLU activation. This has been the case for all datasets, except when $s = 2$ and $3$ for NCI1 (Figure 12 and 13). Since the convex envelop is still a lower bound of the true objective $F_c(A)$, there could be exceptions in some cases for some datasets.

## E.4   More results on PROTEINS

(a) Linear activation

(b) ReLU activation

Figure 15: Fraction of graphs certified robust with $s \in \{2, 3, 4\}$, under robust training (**R**) and non-robust training (**NR**). Dataset: PROTEINS. In (a), all the three lines for NR certified 100% graphs as robust for all $\delta_g$.

(a) $s = 2$

(b) $s = 3$

(c) $s = 4$

Figure 16: Fraction of graphs that are certified as robust (lower green area) and vulnerable (upper red area, percentage $= 100 - y$-axis. **Linear activation.** All are under robust training. Dataset: PROTEINS.

(a) $s = 2$

(b) $s = 3$

(c) $s = 4$

Figure 17: Same as Figure 16, but using **ReLU activation.**

(a) $s = 3$

(b) $s = 6$

Figure 18: Test accuracy under various attacks, and robust training (**R**) or non-robust training (**NR**). ReLU activation. Dataset: PROTEINS.

## E.5 More results on MUTAG

(a) Linear activation      (b) ReLU activation

Figure 19: Fraction of graphs certified robust with $s \in \{2, 3, 4\}$, under robust training (**R**) and non-robust training (**NR**). Dataset: MUTAG.

(a) $s = 2$      (b) $s = 3$      (c) $s = 4$

Figure 20: Fraction of graphs that are certified as robust (lower green area) and vulnerable (upper red area, percentage $= 100 - y$-axis. **Linear activation.** All are under robust training. Dataset: MUTAG.

(a) $s = 2$      (b) $s = 3$      (c) $s = 4$

Figure 21: Same as Figure 20, but using **ReLU activation.**

(a) $s = 3$      (b) $s = 6$

Figure 22: Test accuracy under various attacks, and robust training (**R**) or non-robust training (**NR**). ReLU activation. Dataset: MUTAG.