[Reviews · NeurIPS 2020]

Review 1

Summary and Contributions: The authors formulated a novel lower bound for certified robustness to topological attacks for Graph Convolution Networks (GCNs). Also, the authors introduced a topological attack that outperforms existing topological attacks. The attack is efficient and was applied for the robust training of GCNs. Overall, this work addresses an important problem of certifying GCNs against topological attacks.

Strengths: - The authors extended the convex polytope certificate for topological attacks on graph data against graph convolution networks (GCNs). This is a novel contribution with strong theoretical support. The certificate was derived for one layer and two-layer GCNs with linear and ReLu activation functions. - The authors introduced an approximate attack based on ADMM method, which provides an upper bound on the GCNs robustness. The gap between the lower and upper bound is tight for one-layer GCNs with a linear activation function. Overall, this ADMM attack is the strongest attack on GCNs. - The experimental evaluation confirms the tightness of the introduces bounds. In particular, the bound is tight for linear activation. For relu activations, the gap exists, but their attack provides the strongest upper bound on robustness.

Weaknesses: - It is hard to highlight the limitations of this work for me. Theoretical derivations are sound. However, I could not check the correctness of all proofs. The experimental evaluation is extensive and detailed. - The explanations can be expanded for the readers who are less familiar with GCNs and topological attacks.

Correctness: Yes. However, I was unable to verify the correctness of all proofs because I am not entirely familiar with the area. However, the authors provided all the necessary details for someone with more experience in this area to verify the proofs.

Clarity: The paper is clearly written and easy to follow. At times, the explanations are dense. The author should consider improving the explanations for the readers who are less experienced in this area.

Relation to Prior Work: The paper clearly discusses its relations to the previous work.

Reproducibility: Yes

Additional Feedback: Could the authors explain why the gap between upper and lower bound certificate is larger for linear activation than relu activation? (Example, Figure 11 and 12 in supplementary materials). However, in general, the certificate should be more accurate for linear activation.


Review 2

Summary and Contributions: The authors consider the problem of providing robustness certificates against topology-based attacks on graph convolutional networks. Few other works address the problem of topology-aware attacks on GCNs, and the authors claim that this is the first work that considers attacks that solely focus on the topology (rather than the concomitant features). To provide certificates, the authors consider two approaches: (1) an approach the exactly computes a certificate and (2) an approach that computes a lower bound on the margin between the predicted and next highest class (which can still serve as a certificate). This is similar to the line of work in adversarial robustness that has been address by for example Wong and Kolter. In the first approach, the authors show that sorting algorithms can be used to provide certificates for perturbation sets that allow asymmetrical perturbations WRT to the adjacency matrix. In the second approach, the authors leverage duality and convex envelopes to derive lower bounds on the true robust margin. The authors then verify their results on several experiments.

Strengths: + The formulation of the second method which seeks to find a lower bound on the robust certificate is quite interesting. Indeed, simple Lagrangian duality allows the authors to leverage the preliminaries derived for A^1 and A^2, which can be solved exactly, which was a nice result. + The ADMM formulation is clean, and the steps for solving this discrete problem are nicely described. + The problem seems novel. Most of the work on adversarial robustness has focused on images and deep networks. So this work is quite well-motivated as the field of graph neural networks continues to grow.

Weaknesses: - The details concerning the polar operator and the conditional gradient algorithm were rather sparse. A more intuitive explanation of this polar operator should be provided, despite the fact that it was proposes in a different work. The transition from section 4 to 4.1 was the most difficult part of the paper to parse for this reason. - I was confused by the choice to use only 30% of the total dataset for training and 50% for testing. As the authors mention, the idea is to attack a model on the test set, so they need more data for testing. However, one imagines that a suboptimal model is learned when that much data is held out compared to the models that are deployed in practice. Is it really feasible to learn a good model on 180 graphs (for Enzymes)? [Edit: the experiment in the rebuttal demonstrates that they still achieve impressive performance when the data is split according to the more conventional 80-10-10 split.] - It's unclear to me what the training algorithms are. I suppose that NR (non-robust) refers to empirical risk minimization on (1). What is robust training? I couldn't find what the actual algorithm is here? The authors say something about the hinge loss in reference to equation 1, but then the subsequent optimization using the hinge loss does not involve eq (1); it has to do with the formulation of the robust margin used as the starting point for the robust certificate analysis. Is robust training just using the hinge loss as a suffocate in (1)? This uncertainty makes it hard to interpret the plots. [Edit: the reviewers also address this point, pointing out where they describe this and offering further clarification.]

Correctness: Everything seems correct.

Clarity: The paper is well written, although at times the prose is hard to parse -- particularly around the CG algorithm area (see "weaknesses" section). Small typo (I think). In the related work: "GCNs present a new source of linearity..." --> should it be "nonlinearity" rather than "linearity"?

Relation to Prior Work: The prior work seems to be adequately discussed. Indeed, it seems that there isn't much prior work on attacking a GCN via the topology.

Reproducibility: Yes

Additional Feedback: This is a solid paper. I really like the appeals to duality and the use of ADMM to get an certificate of robustness. The experiments can be improved I think by addressing some of the comments I made above. I am willing to raise my score depending on the authors response.


Review 3

Summary and Contributions: The paper proposed an algorithm to certifying the robustness of GCNs to topological attacks based on Lagrange dualization and convex envelope, which is novel.

Strengths: The paper proposed its method with good theoretical analysis to ensure a tight approximation bounds. Also, the problem discussed in the paper about robustness certification is relevant to NeurIPS community, especially those who are interested in adversarial attacks in graph neural network.

Weaknesses: Basically, this is a strong submission, as for the limitations, maybe the authors should expand the related work part.

Correctness: Yes, the empirical methodology is correct.

Clarity: Yes, the definition and symbols are clearly illustrated.

Relation to Prior Work: This paper discussed the related work in introduction part and illustrated the major difference between its method with others is that it leads to a tight approximation bound. It would be better if the related work is organized by topics relevant to the paper, which makes it friendlier to readers.

Reproducibility: Yes

Additional Feedback:


Review 4

Summary and Contributions: This paper studies the problem of classification for multiple graphs with labels using graph convolutional networks (GCNs) under topological attacks on the graphs. This problem is more complicated than other related graph problems due to the presence of multiple network data, which makes necessary to pull the information of the whole graph, so existing techniques do not apply. The authors study specific instances of a one-layer GCN for classification, and introduce a method to certify robustness to certain topological attacks on altering a number of edges per node or in the whole graph. For certain attacks on the node or graph edges, the authors propose a method that can calculate the certificate exactly using dynamic programming, as well as relaxations of the problem for cases that are computationally infeasible. In addition, the authors introduce an approximate attack using an ADMM algorithm. The certificates and attacks are evaluated on real data, suggesting that these algorithms are tight.

Strengths: The problem studied in this paper is of interest to the community since graph classification is a relevant learning task, and this paper is the first to provide a certificate of robustness for GCNs on this task. The authors provide several algorithms to certify robustness under different attacks and activation functions, and the certificates appear to be tight when compared with an adversarial attack also introduced in the paper (figures 1 and 2), and yields better certificates than other existing methods (fig 3). Moreover, the adversarial attack developed by the authors seem to be more effective for this problem than previous methods (figure 5).

Weaknesses: Update: Thanks to the authors for responding to my concerns. I think that the example provided in the response shows that the method can indeed handle larger graphs. ====================================== I think that one weakness is that the size of the graphs employed in the experiments is very small (median 32 nodes). I wonder whether the method can scale to graphs with larger number of nodes and edges. For example, in MRI applications, the number of nodes is usually in the hundreds or thousands.

Correctness: The derivations of the methodology seem to be correct, and the optimization methods are appropriate to solve the problems.

Clarity: The paper is clearly written in general, and provides motivations that help in understanding the methodology and the problems that are being addressed with the algorithms.

Relation to Prior Work: The authors discuss other related methods for certifying robustness against attacks, and the challenges of using these techniques in graph classification with GCNs. The authors also compare with generic certificates of robustness and attacks in the simulations.

Reproducibility: Yes

Additional Feedback:

[Author Response · NeurIPS 2020]

We wish to thank the reviewers for the comments. We will fix all the issues related to the clarity of presentation.

**Reviewer 1**

**Q:** In dataset NCI1, the gap between upper and lower bound certificate is larger for linear activation than ReLU activation.

**A:** In general, it is true that the gap for linear activation should be smaller than that for ReLU activation. This has been the case for all datasets, except when $s = 2$ and 3 for NCI1 (Figure 11 and 12). We double checked the experiment and did not find an error. Since the convex envelop is still a lower bound of the true objective $F_c(A)$, there could be exceptions in some cases for some datasets.

**Reviewer 2**

**Q:** A more intuitive explanation of polar operator should be provided.

**A:** We will add more details on the conditional gradient algorithm (a.k.a. Frank–Wolfe, `https://en.wikipedia.org/wiki/Frank%E2%80%93Wolfe_algorithm`) and the polar operator (step 1 of the algorithm on the wikipedia page). The projected gradient descent algorithm is not feasible in our context because projection to co($A$) is hard and even explicitly expressing co($A$) in terms of linear constraints can be hard. To bypass this difficulty, the conditional gradient algorithm was adopted, which instead resorts to maximizing a linear function over co($A$) (polar operator). This is equivalent to maximizing a linear function over $A$, and can be solved efficiently as shown in Section 4.1.

**Q:** using 30% of the data for training

**A:** We additionally experimented on Enzyme with 80%, 10%, 10% for training, validation and testing, respectively. The small size of test data led to marked variations in the gap plot, and it is unclear how to "average" them. So we plotted a typical result below, which shows the resulting fraction of certifiably robust / vulnerable for both linear and ReLU activations. Compared with Figure 7 and 8 in the Supplementary material where 30% graphs were used for training, the tightness here appears similar, or slightly better under the linear activation.

(a) Linear ($s = 2$)  (b) Linear ($s = 3$)  (c) ReLU ($s = 2$)  (d) ReLU ($s = 3$)

**Q:** What are the training algorithms? What is the robust training?

**A:** The non-robust (NR) training objective is empirical risk minimization on Eq (1), where $\ell$ is the cross entropy loss. The robust training (R) follows exactly from [11, 12], where a hinge loss (line 264) is added to Eq (1) that encourages a larger prediction margin. A detailed description is provided in the second paragraph of Section 5 (line 263 to 269). Both objectives were optimized by Adam.

**Reviewer 3**

Thank you for your comment. We will improve the organization of related works.

**Reviewer 4**

**Q:** Can the method scale to graphs with a larger number of nodes and edges?

**A:** We examined another dataset DD [1], where the median #node $= 284$ and median #edge $= 716$ per graph. We set $\delta_g = 20$ and $s = 3$. For the certificates with linear activation, Enzyme (median #node $= 32$, median #edge $= 120$) takes 0.37 seconds per graph on average, while DD takes 7.3 seconds. For the certificates with ReLU activation, Enzyme takes 1 second per graph on average, while DD takes 28 seconds. This is consistent with the fact that the computational cost depends quadratically on the number of nodes (in practice, a bit lower than that due to implementation details).

Different datasets have different number of classes. To facilitate comparison, the reported time cost is for each class (see $\min_c$ in line 102.5). The number of edges in the original graph does not affect the computational cost much, because the attacker can both delete **and** add edges.

## Footnotes

[1]Benchmark Data Sets: `https://ls11-www.cs.tu-dortmund.de/staff/morris/graphkerneldatasets`


[Meta-Review · NeurIPS 2020]

The paper proposes an algorithm for certifying the robustness of one layer and two-layer GCNs for graph classification under topological attack. As a byproduct, the authors also propose a new attack algorithm, which, when used in conjunction with the certificate, confirms empirically that both the attack and the certificate are often tight. There is consensus among the four reviewers that the paper should be accepted.